# Spatial pattern of perinatal mortality and its determinants in Ethiopia: Data from Ethiopian Demographic and Health Survey 2016

Tesfaye Assebe Yadeta[1]*, Bizatu Mengistu[2], Tesfaye Gobena[2], Lemma Demissie Regassa[3]

1 School of Nursing and Midwifery, College of Health and Medical Sciences, Haramaya University, Harar, Ethiopia, 2 Department of Environmental Health, College of Health and Medical Sciences, Haramaya University, Harar, Ethiopia, 3 Department of Epidemiology and Biostatistics, School of Public Health, Haramaya University, Harar, Ethiopia

* tesfaye.assb@gmail.com

**Data Availability Statement:** The Demographic and Health Surveys (DHS) 2016 data set is available at https://dhsprogram.com/data/available-datasets.cfm. The DHS Program is authorized to

## Abstract

### Background

The perinatal mortality rate in Ethiopia is among the highest in Sub Saharan Africa. The aim of this study was to identify the spatial patterns and determinants of perinatal mortality in the country using a national representative 2016 Ethiopia Demographic and Health Survey (EDHS) data.

### Methods

The analysis was completed utilizing data from 2016 Ethiopian Demographic and Health Survey. This data captured the information of 5 years preceding the survey period. A total of 7230 women who at delivered at seven or more months gestational age nested within 622 enumeration areas (EAs) were used. Statistical analysis was performed by using STATA version 14.1, by considering the hierarchical nature of the data. Multilevel logistic regression models were fitted to identify community and individual-level factors associated with perinatal mortality. ArcGIS version 10.1 was used for spatial analysis. Moran's, I statistics fitted to identify global autocorrelation and local autocorrelation was identified using SatSCan version 9.6.

### Results

The spatial distribution of perinatal mortality in Ethiopia revealed a clustering pattern. The global Moran's I value was 0.047 with p-value <0.001. Perinatal mortality was positively associated with the maternal age, being from rural residence, history of terminating a pregnancy, and place of delivery, while negatively associated with partners' educational level, higher wealth index, longer birth interval, female being head of household and the number of antenatal care (ANC) follow up.

### Conclusions

In Ethiopia, the perinatal mortality is high and had spatial variations across the country. Strengthening partner's education, family planning for longer birth interval, ANC, and

distribute, at no cost, unrestricted survey data files for legitimate academic research. Registration is prerequisite for access to data. The data sets are publicly available to all registered users, and down loaded from the web site.

**Funding:** The author(s) received no funding for this work.

**Competing interests:** No authors have competing interest.

**Abbreviations:** ANC, Antenatal care; AOR, Adjusted odds ratio; CI, Confidence interval; CSA, Central statistics agency; EA, Enumeration areas; EDHS, Ethiopian demographic and health survey; EPHI, Ethiopian public health institute; FMoH, Federal ministry of health; ICF, International classification of functioning, disability and health; PHC, Population and housing census; SSA, Sub-Saharan Africa.

delivery services are essential to reduce perinatal mortality and achieve sustainable development goals in Ethiopia. Disparities in perinatal mortality rates should be addressed alongside efforts to address inequities in maternal and neonatal healthcare services all over the country.

# Background

Perinatal mortality includes stillbirths and early neonatal deaths [1], and is an important indicator of quality obstetrics and pediatrics care [2]. Perinatal mortality is higher in underprivileged populations showing social inequalities in health care and scarce prenatal services [3]. Sub-Saharan Africa continues to be the region with the highest perinatal mortality rate worldwide at 34.7 deaths per 1,000 live births [4]. Neonatal mortality rate of 27 deaths per 1,000 live births in Sub-Saharan Africa is about 12 times higher than the average in high income countries (2.3 deaths per 1,000 live births) in 2017. Similarly, 23.8 stillbirth per 1,000 live births in Sub-Saharan Africa were about 8 times that found in high income countries (2.97/1000) [5, 6].

Multi-country studies found that low maternal education level [7], alcohol abuse during pregnancy [7], unmarried women [7], overweight or obese mothers [7, 8], home delivery [8], prior preterm delivery [7, 9], spontaneous or induced abortions [7, 9], antepartum hemorrhage [7, 10], fetal growth retardation [7], infections/sepsis [9, 10], pre-pregnancy induced hypertensive disorder [10], and severe anemia [10] were significantly associated with the risk of perinatal mortality.

Most of the causes of perinatal mortality are preventable [11, 12] and available interventions can reduce the three most common causes of neonatal mortality, preterm, intrapartum, and infection related deaths, by 58%, 79%, and 84%, respectively [11]. Improved coverage and quality of preconception, antenatal, intrapartum, and postnatal interventions by 2025 could prevent 71% of neonatal deaths, 33% of stillbirths, and 54% of maternal deaths per year [11].

Ethiopia is among the countries with the highestperinatal mortality in the world with 33 per 1000 live births [13]. There are disparities of maternal and newbornhealth services among different parts of the country. In addition to the difference in health service coverage, the utilization of the available serviceis restricted by low-economic status, illiteracy, unemployment, and feweraccess to mass media [14]. To support the plan, decision, and policiesdevelopment targeted maternal and newborn care, disparities of birth outcomeacross the country is necessary to be established. \Therefore, the objective ofthis study is to analyze spatial patterns of perinatal mortality and itsdeterminants in Ethiopia using the 2016 EDHS national representative data. This studyprovides useful evidence to advance efforts in reducing perinatal mortality toachieve the SDG target plans in ending preventable child deaths by 2030 [1].

# Methods

## Study area and period

Ethiopia whose population is highly diverse, with over 80 ethnic groups, is administratively divided into nine regional states and two city administrations. These are subdivided into 817 administrative Woredas (districts) which are further divided into 16,253 Kebeles, the smallest administrative units of the country (Ethiopia) [15, 16]. The 2007 population and housing census projected that the estimated total population of Ethiopia would be more than 99 million with a total fertility rate of 4.59 by 2015 [17]. More than 80% of the Ethiopian population are

rural inhabitants. There was a mean of 4.6 individuals per household. There has been a marked decline in the total fertility rate from 1990 to 2015, from 6.4 births to 4.6 births per woman in Ethiopia [13]. Ethiopian Demographic and Health Survey is considered as the main data source in the country as it was designed to provide population and health indicators at the national and regional levels. The data collection period was from January 18 to June 27, 2016.

## Data source and study population

This study analyzes EDHS 2016 survey data which available on major DHS program website (https://www.dhsprogram.com/data/dataset_admin). The DHS Program is authorized to distribute, at no cost, unrestricted survey data files for legitimate academic research. Registration is prerequisite for access to data. The data sets are publicly available to all registered users, and down loaded from the web site. This study used the individual recode (IR) data set. The Individual recode dataset has one record for every eligible woman information on socio-demographic, pregnancy, delivery, postnatal care, immunization, and other health related history in the 5 years preceding the survey period. Women who had delivered seven or more month's gestational age, including both stillbirth and live birth and seven days postpartum (o-6 days) in the 5 years preceding the survey period nested within 622 clusters were study participants. A total of 15,683 women of reproductive age were extracted from the data set of whom 7230 women fitting the criteria were used for the analysis. Four hundred forty-five (6.1%) women who had delivered in the 5 years preceding the survey that resides in 23 clusters were omitted from the spatial analysis since they had no geographic information.

## Sampling methods

The sampling frame used was Ethiopia Population and Housing Census (EPHC) conducted in 2007 by the Ethiopian Central Statistical Agency. The census frame conducted in 2007 by the Ethiopia Central Statistical Agency (CSA) is a complete list of 84,915 enumeration areas (EAs). Enumeration areas are a geographic area covering on average 181 households [15, 16]. In the 2016 EDHS, the sample was stratified and selected in two stages [13]. Each region was stratified into urban and rural areas, yielding 21 sampling strata. Samples of EAs were selected independently in each stratum in two stages. Implicit stratification and proportional allocation was achieved at each of the lower administrative levels by sorting the sampling frame within each sampling stratum, before the sample selection, according to administrative units in different levels, and by using a probability proportional to size selection during the first stage of sampling.

In the first stage, a total of 645 EAs, 443 in rural and 202 in urban areas were selected with probability proportional to EA size and with independent selection in each sampling stratum. In the second stage of selection, a fixed number of 28 households per cluster were selected with an equal probability systematic selection from the newly created household listing. All women age 15–49 who were either permanent residents of the selected households or visitors who stayed in the household the night prior to the data collection day were eligible for interview.

Geographic coordinates of each survey cluster were also collected using Global Positioning System (GPS) receivers. The GPS reading was collected at the center of each cluster. To ensure confidentiality, GPS latitude/longitude positions for all surveys were randomly displaced before public release. The maximum displacement was two kilometers for urban clusters and five kilometers for 99% of rural clusters. The one percent of rural clusters were displaced a maximum of 10 km. The displacement was restricted within the country's second administrative level [15, 18].

## Survey tool

The DHS questionnaires were adapted from model survey instruments developed for the measure DHS project to reflect the population and health issues are relevant to Ethiopia. The adaptation of the questionnaire was conducted through a series of meetings with the various stakeholders. In addition to the English language, the questionnaires were translated into three major local languages; Amharigna, Afaan Oromo, and Tigrigna. The Woman's questionnaire was used to collect information from all women of reproductive age (15–49 years).

## Measurement

### Outcome measures

The perinatal mortality rate was outcome variable, defined as stillbirths plus early neonatal deaths in the five years preceding the survey divided by all births (including stillbirths) that had a pregnancy duration of 7 or more months. Stillbirth was defined as the number of fetal deaths in pregnancy of seven or more months [13]. An early neonatal death is defined as a death in the first seven days (days 0–6) of a child born alive [1]. The details of perinatal mortality rate, stillbirth, and early neonatal death calculation method found on the DHS Contraceptive Calendar tutor [19]. Dummy variables were created for this variable for presence of perinatal mortality and was assigned "yes" (coded as 1) and the absence of perinatal mortality was coded as "No" (coded as 0). Thus, the outcome variable for the $i^{th}$ the child is dichotomous, represented by a random variable $Y_{ij}$ that takes the value "1" with a probability of success (perinatal death) and the value "0" with probability of failure (no perinatal death), such that;

$$Yij = \begin{cases} 1 \ if \ there \ is \ perinatal \ death \\ 0 \ if \ there \ is \ no \ perinatal \ death \end{cases}$$

When Y is the outcome variable, while i is for the child and child-level factors, while j is for the community factors.

### Exposure measures

Potential predictors of perinatal mortality such as socio-demographic characteristics including age, mothers' educational level, occupation and wealth index, health service factors including antenatal care (ANC), tetanus toxoid (TT) vaccination status, exposure to health service message and health insurance utilization, age at marriage and age at first birth were considered as independent variables and included in the analysis. Community-level variables included in the analysis were a place of residence (urban, rural), distance from health facilities and regions.

### Statistical analysis

A STATA format of women reproductive age group EDHS 2016 data set was downloaded. All the statistical analysis was conducted by using STATA software version 14.1. Sample weights were applied to compensate for the unequal probability of selection between the strata that had previously been geographically defined. A detailed explanation of the weighting procedure can be found in the methodology of the 2016, EDH Survey final report. Hence, considering the hierarchical nature of the data, multilevel logistic regression models were fitted to identify community and individual level factors associated with perinatal mortality by using "xtmelogit" code of STATA. To test the significance of the variance of the random intercept, the likelihood ratio test was applied. Measures of variation for random effect were determined by

computing the intraclass correlation coefficient (ICC), median odds ratio (MOR) [20] and proportional change in variance (PCV) statistics [21, 22]. The intraclass correlation coefficient is a measure of within-cluster variation (i.e. the variation between individuals within the same cluster) [23, 24]. The PCV is a measurement of the total variation attributed to individual and / or community-level factors at each model. The MOR is the median odds ratio between the individual of higher propensity and the individual of lower propensity when comparing two individuals from two different randomly chosen clusters and it measures the unexplained cluster heterogeneity (the variation between clusters) by comparing two persons from two randomly chosen different clusters. The MOR measure is always greater than or equal to "1". If The MOR measure is "1", there is no variation between clusters. The within-cluster correlation was measured using intracluster correlation (ICC) which is expected to be$\geq$ 10% to use the model.

The formulas for these 3 measurements are as follows;

ICC = CLUSTER LEVEL VARIANCE / TOTAL VARIANCE i.e. $ICC = \frac{v_i}{v_i + \frac{\pi^2}{3}}$, where

vi = estimated variance in each model, which has been described elsewhere [25].

$PCV = \frac{v_x - v_y}{v_x}$, $v_x$ = variance of the initial model, and $v_y$ = variance of the model with more terms.

MOR = exp(sqrt(2*CLUSTER LEVEL VARIANCE))*invnormal(0.75)) i.e. $MOR = exp(0.95\sqrt{v_z})$ if $v_z$ = is the cluster level variance.

Adjusted odds ratio with a 95% confidence level was reported to show the strength of the association and it's a significance for the fixed effect. Variables having p-value < 0.05 was considered as having significant association with the outcome. The model goodness of fit was checked using deviance information criteria (DIC) and Akaike information criteria (AIC) and the model with the lowest value was considered to be the best fit model.

## Spatial analysis

The spatial analysis helps to show cluster and regional variation in perinatal mortality and reveals abnormal patterns. Using the geospatial method aids in faster and better health mapping and analysis than conventional methods. The spatial analysis was carried out using ArcGIS 10.1 and Sat Scan 9.6. To produce the flattened map of Ethiopia, the Ethiopian Polyconic Projected Coordinate System was used [26]. For the analysis, the aggregated perinatal mortality count data was joined to the geographic coordinates based on each EA unique identification code. To evaluate whether the pattern expressed is clustered, dispersed, or random across the study areas, global spatial autocorrelation was assessed with ArcGIS using the Global Moran's I statistic (Moran's I) [26]. When p-value indicates statistical significance, a positive Moran's I index value indicates a tendency toward clustering while a negative Moran's I index value indicates a tendency toward dispersion [27].

We employ the logarithm of the likelihood ratio (LLR) as the test statistic to identify maximum likelihood clusters. By considering the detected cluster ($\hat{Z}$) is the maximum the likelihood estimator of $Z$ maximum likelihood estimation method is also applied to determine the most clustered sub-region $Z$. If $C$ and cz be the observed number of perinatal mortality in $G$ and $z$, respectively, whereas $N$ and $n_z$ are the expected number of perinatal mortality in $G$ and $z$ under the null hypothesis; hence, N = C. Let $L(z)$ be the likelihood under the alternative hypothesis that $z$ is a cluster and $L_0$ be the likelihood under the null hypothesis; LLR is [28]:

$$\frac{L(z)}{Lo} = \left(\frac{cz}{nz}\right)cz\left(\frac{C - cz}{C - nz}\right)C - cz$$

Hence the final LLR of Z cluster is:

$$LLR(z) = ln\frac{L(z)}{Lo}$$

Where: $L_0$ is a constant for a given $G$. The collection $z$ of spatial units can maximize $LLR(z)$ and $L(z)$. In the presence of positive global spatial autocorrelation, local spatial clusters of areas with high or low perinatal mortality was detected using Kulldorff's method of purely spatial scan statistic assuming the discrete Poisson probability model in SaTScan [29]. To avoid the detection of large clusters, we used a maximum of 10% of the population at risk for the spatial cluster size and the analysis was done using standard Monte Carlo hypothesis testing with 999 Monte Carlo replicates [30, 31].

A cluster is statistically significant when its log likelihood ratio (LLR) is larger than the Standard Monte Carlo critical value (C.V) for 0.05 significance level or p-value < 0.05.

## Ethics approval and consent to participate

Ethical approval of EDHS was obtained from the ICF Institutional Review Board (IRB), Ethiopia Health and Nutrition Research Institute Review Board, and the Ministry of Science and Technology. The data collectors read the informed consent statement to obtaining informed and voluntary participation before data collection. The confidentiality of the information was maintained. For this particular study, a brief description of the protocol was submitted to the MEASURE DHS program to access and analyze the data. Permission was obtained from the program to access and analyze the data.

## Results

### Perinatal mortality by socio-demographic characteristics

Of 7230 total study participants, 7180 were live births, 235 were early neonatal deaths and 115 were stillbirths. The median age of the respondents was 28 years and the partner's age range was from 15 to 95, with a median age of 37 years. Of the total, 5713 (79%) where rural resident, and 4384 (61%) had no formal education. Perinatal mortality rate was high among rural residents (32.0 per 1,000 birth), and among respodents with no education (33 per 1,000 birth). Perinatal mortality was only 61 (35 per 1000 birth) among households headed by a female (Table 1).

### Perinatal mortality by reproductive health related characteristics of mothers

The median age at first birth was 19 years ranged between 11 and 40 years, the median length between births was 42 months. Only 682 (9.43%) mothers had history of terminating pregnancy. Of 7230 pregnant women, 3,019 (41.26%) had at least 2 doses of tetanus toxoid (TT) vaccine during pregnancy, 4683 (64.77%) had at least one ANC follow up, 2,694 (37.52%) were gave birth at institutional, 675(9.40%) had postnatal care. Regarding health insurance, 258 (3.57%) had accessed to community-based health insurance service. More than halve (53%) of the respondents reported problem of accessing health service (Table 2).

### Perinatal mortality rate and spatial destribution

Overall the perinatal mortality rate was 33 per 1000 pregnancy. Perinatal mortality rate was varied over the regions ranging from 50 per 1000 pregnancy in Somali region to 24.5 per 1000 pregnancy in Afar region (Fig 1).

**Table 1. Perinatal mortality rate by socio demographic characteristics of mothers who were pregnant and give birth during 5 years preceding the 2016 EDHS survey.**

| Variable | Categories | Total birth (%) | Perinatal mortality rate (PMR per 1000) |
|---|---|---|---|
| Age in year | Less than 20 | 359 (4.97) | 12 (20.93) |
| | 20–34 | 5,063 (70.03) | 225 (29.74) |
| | 35–44 | 1,634 (22.60) | 96 (42.75) |
| | 45–49 | 174 (2.41) | 17 (59.86) |
| Residence | Rural | 5,713 (79.02) | 294 (32.0) |
| | Urban | 1,517 (20.98) | 56 (16.0) |
| Women education | No education | 4,384 (60.64) | 233 (32.84) |
| | Primary | 1,953 (27.01) | 85 (32.36) |
| | Secondary | 575 (7.95) | 21 (33.01) |
| | Beyond secondary | 318 (4.40) | 11 (51.69) |
| Husband Education | No education | 3,197 (47.79) | 183 (32.18) |
| | Primary | 2,176 (32.53) | 115 (33.93) |
| | Secondary | 745 (11.14) | 31 (28.27) |
| | Post-secondary | 572 (8.55) | 21 (39.79) |
| Sex of household head | Male | 5,626 (77.81) | 289 (32.73) |
| | Female | 1,604 (22.19) | 61 (35.0) |
| Number of children in the house | 0 | 336 (4.65) | 103 (27.51) |
| | 1–3 | 6,781 (93.79) | 244 (25.26) |
| | 4–6 | 113 (1.56) | 3 (11.23) |
| Wealth index | Poorest | 2,487 (34.40) | 136 (27.44) |
| | Poorer | 1,348 (18.64) | 61 (28.15) |
| | Middle | 1,237 (17.11) | 53 (34.78) |
| | Richer | 1,103 (15.26) | 42 (39.28) |
| | Richest | 1,055 (14.59) | 58 (39.74) |

PMR;Perinatal Mortality Rate.

## Global Moran's I test of cluster distribution

The spatial distribution of perinatal mortality in Ethiopia was non-random as the global spatial autocorrelation analysis based on feature locations and attribute values revealed a clustering pattern of perinatal mortality across the study areas. The global Moran's, I value was 0.047 with P-value <0.001 (Fig 2).

## Geographical variation of perinatal mortality rate

The global spatial autocorrelation of perinatal mortality based on feature locations and attribute values across the study area, revealed regional variation. When the results were subdivided by zone, prenatal mortality rate was low in Addis Ababa, some parts of Tigray and Gambela regions. Somali region had high perinatal mortality rates (Fig 3).

## Local spatial clusters area analysis

Local spatial clusters of areas with high or low perinatal mortality detected. The red window indicates the identified significant clusters inside the window. Twenty-nine most likely clusters were identified in spatial scan statistics which were at 7.829198 N, 43.706264 E with 206.18 km radius (in the Somali region of Ethiopia). The risk of perinatal mortality was three times higher among mothers living in the primary cluster as compared to outside the window (RR = 3.04, LLR = 16.92, P-value <0.001) (Fig 4).

**Table 2. Perinatal mortality by reproductive health related characteristics during the 5 years preceding the 2016 EDHS survey.**

| Variables | Categories | Total birth (%) | Perinatal mortality (PMR per 1000) |
|---|---|---|---|
| Age at first birth | Less than 18 years | 2,686 (37.15) | 136 (34.0) |
| | Between 18–35 years | 4,535 (62.49) | 197 (28.29) |
| | After 35 | 26 (0.36) | 17 (38.80) |
| Birth interval | Less than 15 | 260 (4.58) | 189 (97.01) |
| | 15–30 | 1,901 (33.50) | 109 (26.30) |
| | More than 30 | 3,513 (61.91) | 52 (31.71) |
| Pregnancy ever terminated | No | 6,548 (90.57) | 207 (21.66) |
| | Yes | 682 (9.43) | 143 (13.75) |
| Health insurance (CBHI) | No | 6,972 (96.43) | 339 (32.81) |
| | Yes | 258 (3.57) | 11 (39.62) |
| Numbers of ANC follow up | Not visit at all | 2,547 (35.23) | 182 (29.0) |
| | 1–3 | 2,089 (28.89) | 73 (24.53) |
| | 4 and above | 2,594 (35.88) | 95 (46.80) |
| TT-vaccine (during pregnancy) | Not at all | 3,214 (44.76) | 127 (26.84) |
| | Once | 834 (11.62) | 53 (29.24) |
| | 2 and more | 2,983 (41.26) | 169 (29.58) |
| | Unknown | 149 (2.06) | 1 (17.41) |
| Place of delivery | Home | 4,486 (62.48) | 190 (24.37) |
| | Health Institution | 2,694 (37.52) | 160 (52.22) |
| Skilled birth attendant | No | 4,399 (61.27) | 215 (32.16) |
| | Yes | 2,781 (38.73) | 135 (34.83) |
| Postnatal service | No | 6,500 (90.53) | 294 (28.83) |
| | Yes | 675 (9.47) | 56 (14.03) |
| Distance from Health facilities | Big problem | 3,826 (52.92) | 192 (31.15) |
| | Not a big problem | 3,404 (47.08) | 158 (36.0) |

PMR; Perinatal Mortality Rate, TT; tetanus Toxoids, ANC; Antenatal care, CBHI; Community Based Health Insurance.

## Factors associated with perinatal mortality

The final best fitted model (Table 3) was used for the determination of associated factors with the perinatal mortality. Hence, from the final model, the risk of perinatal mortality was raised with the increment of mother's age. The odds of perinatal mortality was higher by 41% (AOR = 1.41, 95%CI: 1.1, 1.87) among 35–44 age groups and twice (AOR = 2.06, 95%CI: 1.13–3.78) among 45–49 when compared to mothers aged 20–34 years. The odds of perinatal mortality were lowered by 26% (AOR = 0.74, 95%CI: 0.49, 0.98) among mothers whose partners level of education was higher education when compared to those whose partners had no education. Being from a household in which head of the household was female lower the odds of perinatal death by 37% (AOR = 0.63, 95% CI: 0.45, 0.88) compared to those households with males as heads of households. The odds of perinatal mortality were lowered by 41% (AOR = 0.59, 95% CI 0.38, 0.93) in households with highest wealth index as it compared to the poorest households (Table 3).

Birth interval lower the odds of perinatal mortality by 70% (AOR = 0.30, 95%CI: 0.21, 0.45) as the distance between birth was between 15 and 30 months and 82% (AOR = 0.18, 95%CI: 0.12, 0.27) when distance between birth was more than 30 months as compared to less than 15 moths. History of terminating a pregnancy was related to the increment of odds of PNM by 65% (AOR = 1.65, 95%CI: 1.32–1.97) as compared to their counterpart (Table 3).

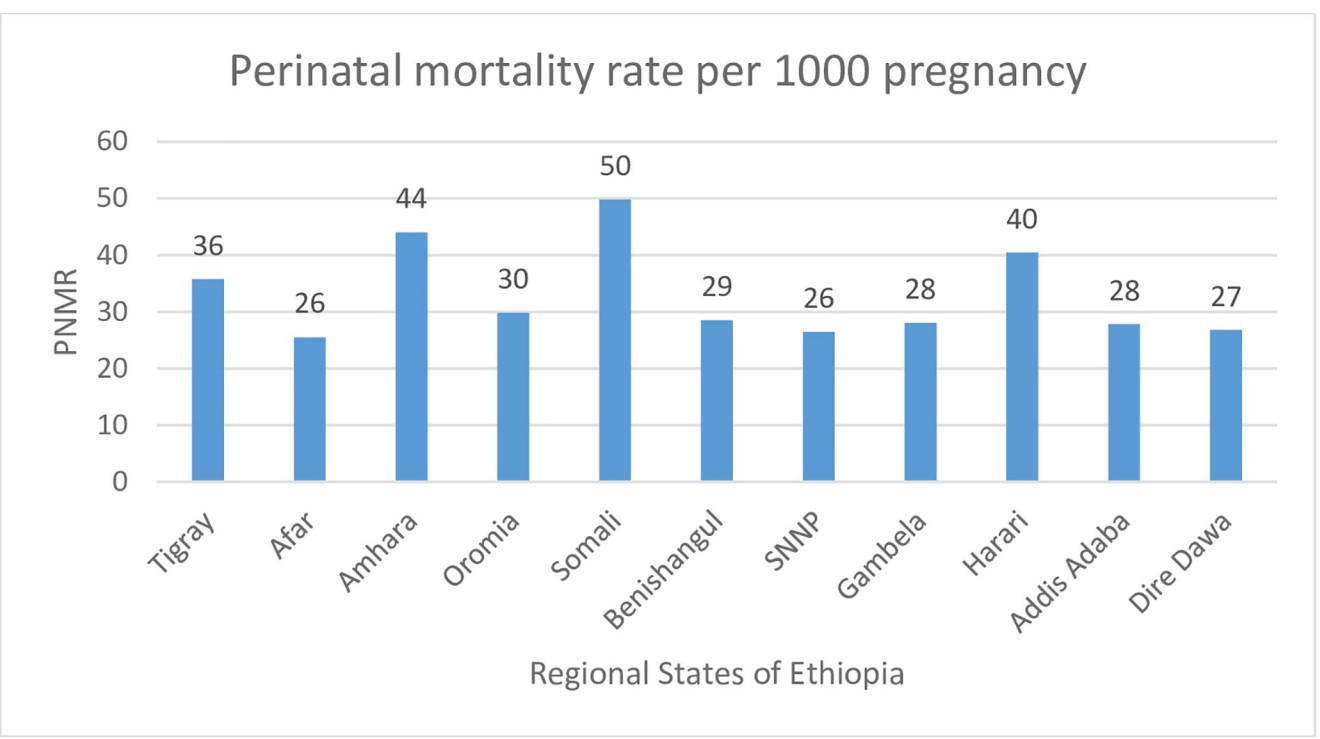

**Fig 1. Perinatal mortality rate among each regional states of Ethiopia, EDHS 2016.**

The odds of PNM was lower by 63% (AOR = 0.37, 95%CI: 0.25–0.55) among mothers who had at most three ANC follow up the history and by 46% (AOR = 0.54,95%CI: 0.36–0.82) among those who had at least four when compared to mothers who had no history of ANC follow up. The odds of PNM was about two times higher (AOR = 1.83, 95%CI: 1.28–2.63) among mothers who gave birth a home compared to health institutional delivery (Table 3).

The community-level variables (residence and region) also showed significant association in the final model. The odds of perinatal mortality among rural resident was twice (AOR = 2.28, 95%CI: 1.25, 4.17) that of urban the residents. The odds of perinatal mortality was about four times higher (AOR = 3.62, 95%CI: 1.72, 7.61) among mothers from the Somali region and twice higher (AOR = 2.33, 95%CI: 1.08, 5.04) among mothers from the Harari region when it compared to mothers from Addis Ababa (Capital City) (Table 3).

### Measures of variation (random-effects) and model fit statistics

As the results of multilevel logistic regression analysis indicated in Table 4, the null model (Model I) revealed statistically significant variation in perinatal mortality across communities [$\tau$ = 0.69, p = 0.01], in which around 18% variation in the odds of a perinatal death is attributed to community-level factors (ICC = 17.6% (Table 4).

After adjusting the model for household-related factors (model II), the variation in the odds of perinatal mortality remained statistically significant [$\tau$ = 0.57, p < 0.001] across the communities, with 67% of variation in the odds of perinatal mortality was attributed to the household level factors and 15% of the variance in perinatal death was attributed to community-level factors (ICC = 14.8%) (Table 4).

Model III, which was adjusted for community-level factors, revealed lower variance for a perinatal death [$\tau$ = 0.29, p < 0.001] across the communities, as compared to the variance

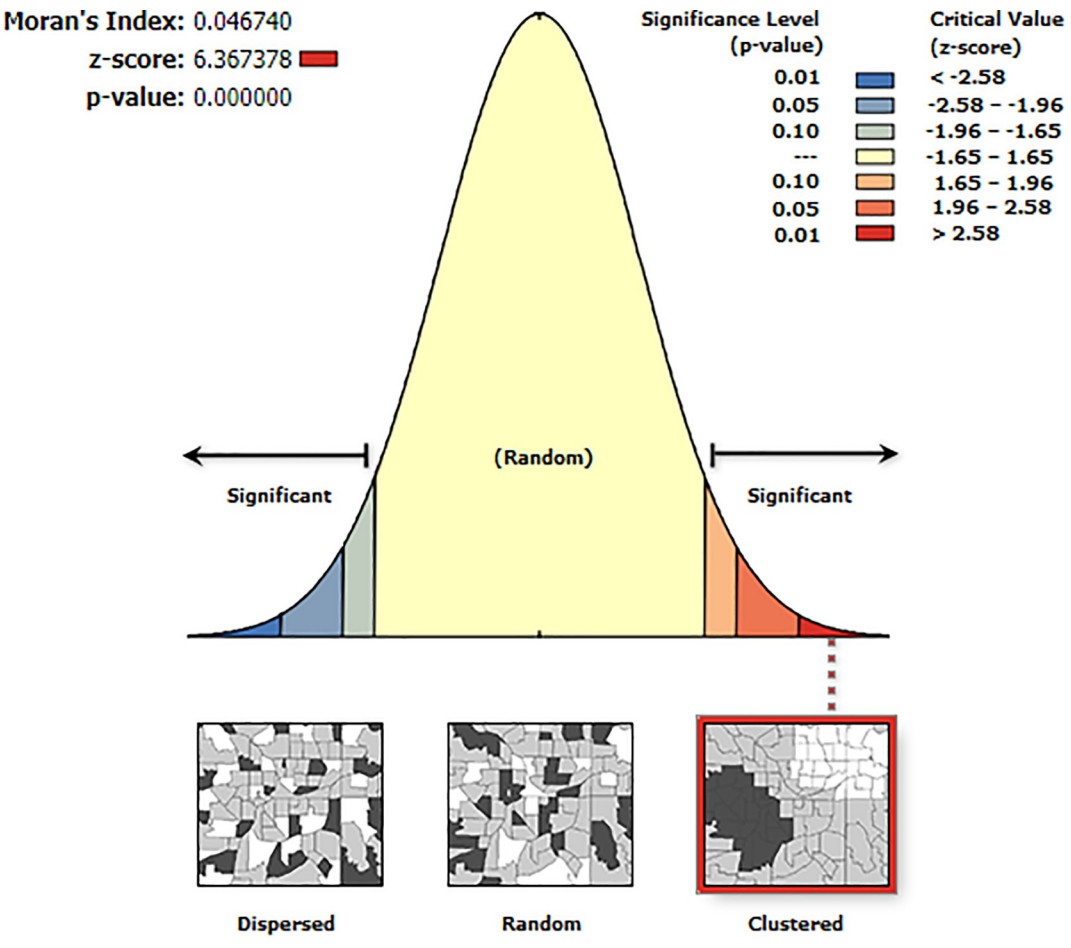

Given the z-score of 6.37, there is a less than 1% likelihood that this clustered pattern could be the result of random chance.

**Fig 2. Global Moran's I test of cluster distribution for perinatal mortality in Ethiopia, from EDHS 2016.**

reported in the model for household-level factors (model II). In this the model, the community-level factors explained the 58% of the variability in the odds of perinatal death (PCV = 57.69%), and 26% of the variation among the clusters were attributed to community level factors (ICC = 25.62%) (Table 4).

The final best-fit model, model IV (AIC = 2413.37), was adjusted for both household and community-level factors simultaneously showing statistically significant variability to the odds of perinatal death [$\tau$ = 0.55, p < 0.001] across the community and household level. This model indicated about 14% of the variability among communities in the odds of perinatal death was due to the factors at the community level (ICC = 14.3%) and about 70% of the variance in the odds of perinatal death (PCV = 70.3%) in the country was attributed to both household and community-level factors (Table 4).

The MOR shows the extent to which the perinatal mortality was determined by residential area and is therefore appropriate for quantifying contextual phenomena. It quantifies the variation in perinatal death between clusters by comparing two mothers with the history of pregnancy within the 5 years preceding the survey from two randomly chosen, different clusters.

**Perinatal mortality rate in EAs per 1,000 population**

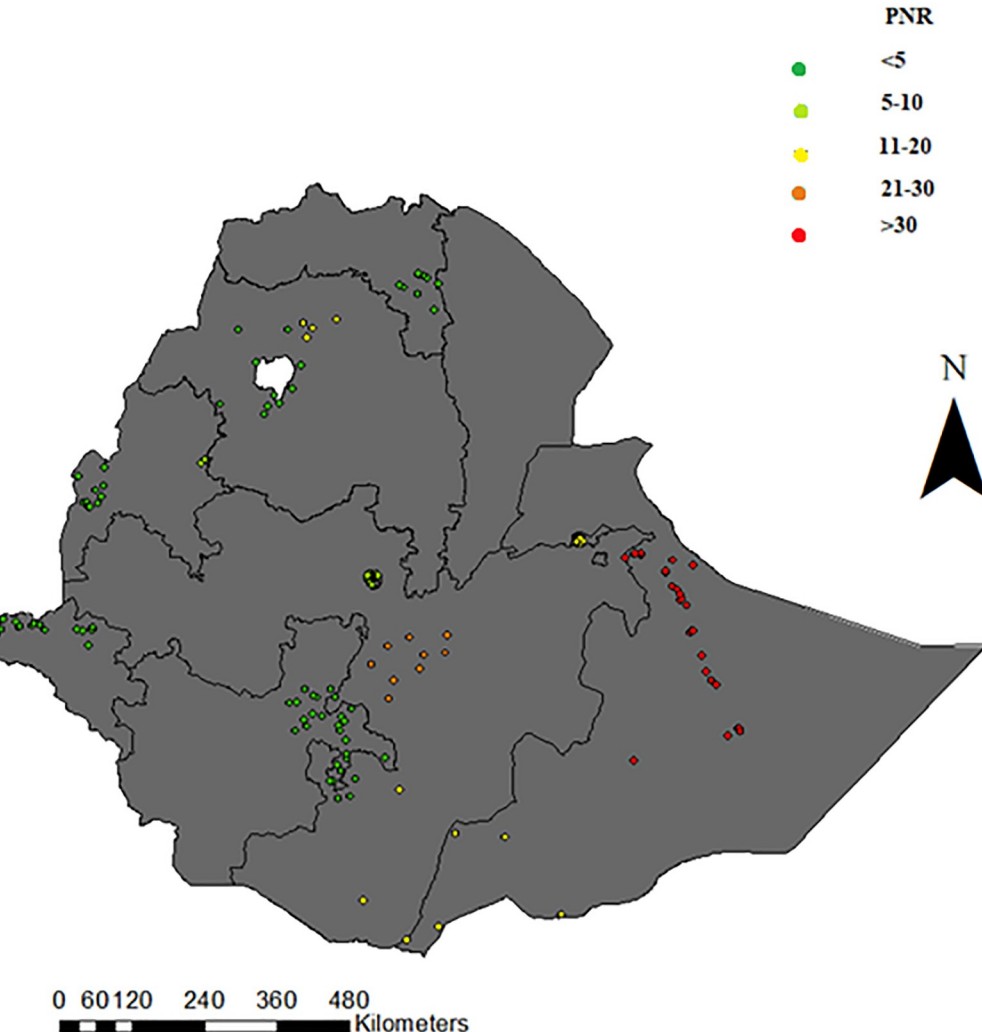

**Fig 3. Geographical variation of perinatal mortality rate of Ethiopia, from EDHS 2016.**

MOR greater than 1 in all models suggests a considerable between-cluster variation in perinatal death. Including both individual, and community-level factors slightly reduced the unexplained heterogeneity in perinatal death between communities from MOR of 2.21 in the null model to the MOR of 1.67 in the 2nd model (community level) and to 1.80 in the final model. Random effect and model fitness statistics for models are summarized in (Table 4).

## Discussion

The study showed variation of perinatal mortality among different geographic locations and clusters. This large survey also revealed that ANC utilization, institutional delivery, increasing the gap between births, increased family wealth and partner education were significantly associated with reduced perinatal mortality.

There were perinatal mortality variations among clusters in this study. The finding was in line with previous studies done elsewhere [32, 33]. This may be due to high disparity in

## Most likely SatScan cluster areas with high and low PMR

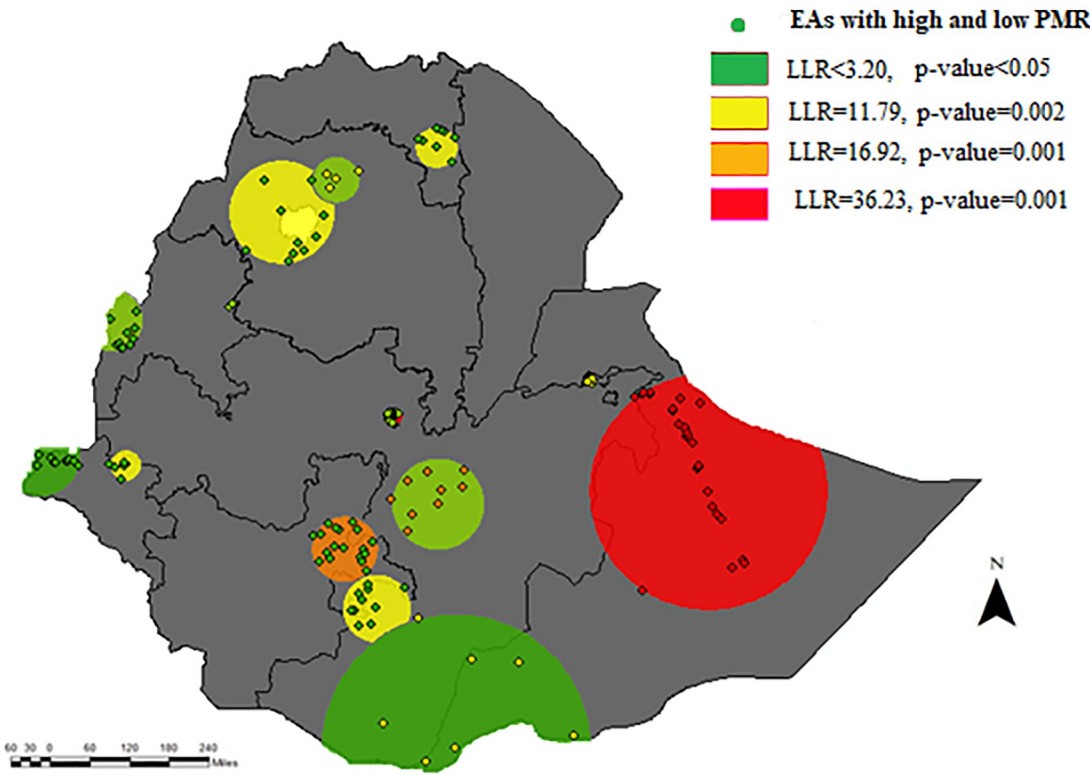

**Fig 4. Perinatal mortality rate possible clusters identified by SAT SCAN, from EDHS 2016.**

maternal and neonatal health services among geographic locations [14]. The increased mortality risk in the Somali region may be explained by the inequity in access to healthcare services, and delay due to long travelling distances to health facilities. The nature of unstable settlement and nomadic pastoralists also poses challenges in accessing healthcare services [34]. In addition, rural residence were associated with perinatal mortality which was also found in the study in Nigeria [35]. Rural residents often encounter barriers to healthcare that limit their ability to obtain basic healthcare needs. Even when healthcare services exists in the community, other barriers exists including lack of financial means to pay for services, transportation, and the ability to take paid time off of work to use such services [36].

This study revealed husband education was associated with perinatal mortality and this was similar to a study done in Bangladesh [37]. Literate men were twice more likely to have positive health-behavior perceptions [38]. The risk of neonatal mortality was low when headed by an educated household [39]. Providing safe motherhood education to husbands resulted in further improvement of some indicators [40]. Husbands are often the decision-makers when it comes to seeking medical care for obstetric complications, or to arrange for transportation.

Our study shows that a women as head household were less likely to experience perinatal mortality. A similarly study from Nepal revealed, women who were from female-headed households were less likely to see a child die than were women in their comparison group [41]. A possible explanation might be women who have autonomy in decision makings are more

Table 3.  Multilevel mixed effect logistic regression for perinatal mortality and associated factor in EDHS 2016.

| Variables | Categories | Model II (AOR) | Model III (AOR) | Model IV (AOR) |
|---|---|---|---|---|
| Household level characteristics | | | | |
| women age in years | 20–34 | 1 | | 1 |
| | Less than 20 | 0.64 [0.15–2.82] | | 0.62 [0.14–2.75] |
| | 35–44 | 1.44 [1.08–1.918] * | | 1.41 [1.1–1.87] * |
| | 45–49 | 2.26 [1.23–4.14] * | | 2.06 [1.13–3.78] * |
| Women education | No-education | 1 | | 1 |
| | Primary | 0.77 [0.58–1.02] | | 0.62 [0.48–1.45] |
| | Secondary | 0.51 [0.31–0.81] * | | 0.93 [0.69–2.24] |
| | Post-secondary | 0.55 [0.29–1.04] | | 0.95 [0.36–2.48] |
| Husband/ partner education | No-education | 1 | | 1 |
| | Primary | 0.96 [0.75–1.24] | | 1.01 [0.84–1.44] |
| | Secondary or above | 0.60 [0.42–0.85] * | | 0.74 [0.49–0.98] * |
| Head of household | Male | 1 | | 1 |
| | Female | 0.68 [0.49–0.94] * | | 0.54 [0.40–0.72] * |
| Wealth index | Poorest | 1 | | 1 |
| | Poorer | 0.83 [0.59–1.19] | | 0.863 [0.60–1.23] |
| | Middle | 0.83 [0.57–1.20] | | 0.88 [0.60–1.29] |
| | Richer | 0.69 [0.46–1.03] | | 0.76 [0.50–1.14] |
| | Richest | 0.48 [0.31–0.74] * | | 0.59 [0.38–0.93] * |
| Birth interval in months | Less than 15 | 1 | | 1 |
| | 15–30 | 0.29 [0.19–0.43] * | | 0.30 [0.21–0.45] * |
| | More than 30 | 0.17 [0.11–0.25] * | | 0.18 [0.12–0.27] * |
| Ever had terminated pregnancy | No | 1 | | 1 |
| | Yes | 1.64 [1.3–1.96] * | | 1.65 [1.32–1.97] * |
| Health insurance[+] | No | 1 | | 1 |
| | Yes | 1.09 [0.56–2.13] | | 1.01 [0.51–1.97] |
| ANC follow up | Not at all | 1 | | 1 |
| | Twice or less | 0.38 [0.26–0.55] * | | 0.37 [0.25–0.55] * |
| | More than 2 | 0.50 [0.34–0.735] * | | 0.54 [0.36–0.82] * |
| TT-vaccination | No | 1 | | 1 |
| | Yes | 0.89 [0.80–0.98] * | | 0.91[0.81–1.01] |
| Place of delivery | Institution | 1 | | 1 |
| | Home | 1.78 [1.25–2.54] * | | 1.83 [1.28–2.63] * |
| Community level characteristics | | | | |
| Distance from HF | Not-big problem | | 1 | 1 |
| | Big problem | | 1.26 [0.99–1.59] | 1.17 [0.89–1.54] |
| Residence | Urban | | 1 | 1 |
| | Rural | | 2.63 [1.80–3.84] * | 2.28 [1.25–4.17] * |
| Region | Addis Ababa | | 1 | 1 |
| | Tigrai | | 1.50 [0.70–3.20] | 1.50 [0.70–3.23] |
| | Afar | | 1.24 [0.55–2.80] | 1.21 [0.53–2.80] |
| | Amhara | | 1.71 [0.80–3.64] | 1.57 [0.73–3.39] |
| | Oromia | | 1.45 [0.68–3.10] | 1.32 [0.61–2.85] |
| | Somali | | 3.77 [1.82–7.78] * | 3.62 [1.72–7.61] * |
| | Benishangul | | 1.44 [0.65–3.20] | 1.28 [0.57–2.88] |
| | SNNP[υ] | | 1.04 [0.49–2.36] | 0.95 [0.43–2.09] |
| | Gambela | | 1.20 [0.52–2.76] | 1.17 [0.51–2.70] |

(*Continued*)

**Table 3.** (Continued)

| Variables | Categories | Model II (AOR) | Model III (AOR) | Model IV (AOR) |
|---|---|---|---|---|
| | Harari | | 2.68 [1.25–5.76] | 2.33 [1.08–5.04] * |
| | Dire Dawa | | 1.01 [0.42–2.40] | 0.92 [0.38–2.22] |

Model I: is Null model without predictors, model II: model fitted for household-related factors, Model III: model fitted for community level factors, Model IV: final model fitted for both household-related and community level factors.

+ = health insurance is the community-based health insurance (CBHI)

* = p<0.05

υ = South Nation Nationalities and people region, AOR = Adjusted Odds Ratio.

likely to use contraceptive, and facility care which reduce the reproductive health risk in perinatal mortality reduction [42].

This study showed an association between perinatal mortality and age as the odds of PNM increased with an advanced age. Studies in the UK [43], Gaza and West Bank [44] and Uganda [45] were reported similar results. Though the exact the mechanism underlying the pathogenesis of perinatal death among older mothers is unclear, myometrial under perfusion caused by sclerotic arterial lesions [46] and aging endothelium women of advanced age might be responsible for perinatal complications [47]. Previous evidence also indicated, pregnancy-induced hypertension, gestational diabetes and stillbirths were prevalent more often in older pregnant women [47–49]. Women advanced age were at increased risk of stillbirth, pre-term birth, macrosomia and intrapartum asphyxia which all may be linked with perinatal death [43, 44].

This study revealed that lower socio-economic factors contributed to increased rates of adverse outcomes [50]. Similarly, this study showed perinatal mortality was decreased by 41% in households with the highest wealth index as compared to the poorest households. The association was also seen in the study conducted in Brazil [51] and Tigray region of Ethiopia [52].

In the present study, the birth interval was one of the predictors of perinatal mortality. We found the birth interval greater than 15 months were significantly associated with decreased risk of perinatal mortality and similar findings were reported in other studies conducted in a different parts of the world [36, 53, 54]. The reason suggested by different literature for the effect of birth interval was maternal depletion if the gap after preceding birth was too short [55, 56]. The frequent pregnancies and periods of lactation leads to a deterioration in the mother's nutritional status and this leads to the risk of subsequent fetal loss, premature birth, and intrauterine growth retardation. These factors result in the increment risk of higher perinatal mortality (stillbirth and early neonatal death) [55, 57]. Avoidance of short intervals can be achieved through the postpartum provision of contraception [58, 59]. To achieve the sustainable development goal to and end all preventable perinatal mortality, health promotion

**Table 4. Random effect parameters and model fitness statistics for multilevel mixed effect logistic regression (models).**

| Estimation | Model I (null model) | Model II (household-related factors) | Model III (community level) | Model IV (both community and individual level) |
|---|---|---|---|---|
| Variance* | 0.69(0.10) | 0.57(0.13) | 0.29 | 0.55 (0.17) |
| ICC% | 17.6 | 14.8 | 25.62 | 14.3 |
| MOR | 2.21 | 2.05 | 1.67 | 1.24 |
| PVC% | 1(reference) | 67 | 57.69 | 70.3 |
| DIC | 3,275.65 | 2,405.73 | 3,157.32 | 2,361.37 |
| AIC | 3279.65 | 2428.60 | 3185.32 | 2413.37 |

programs must better target those communities where contraceptive utilization is low. Urgent effective interventions are therefore needed to improve the strategy of prolonged birth intervals.

The odds of PNM were increased among the women who had a history of terminating the pregnancy. This finding is similar to the study conducted in the Germany [60], Tanzania [61] and Australia [62]. The association between perinatal mortality and history of abortion is unclear but it is evident that previously threatened abortion is significantly associated with a number of complication in subsequent pregnancy outcomes such as preterm delivery, placenta retention, antepartum and postpartum hemorrhage, low birth weight and other causes of still-birth or early neonatal death [63, 64].

Maternal health service utilization (antenatal care and place of delivery) are the major factors that significantly reduce the odds of death during the perinatal period. Hence, the odds of death during perinatal period were significantly reduced among women who had access to and utilized ANC, and gave birth at health facilities attended skilled birth attendants. These significant associations were found in studies conducted elsewhere [65–67]. This could be due to health service utilization during the pregnancy increases the maternal literacy, early awareness of pregnancy complications and better self-care [68]. Additionally, home delivery could increase the early neonatal death by increasing the risk of asphyxia, sepsis, and chorioamnionitis [65, 69]. The Health Extension Program launched to deliver mostly disease prevention and health promotion activities in Ethiopia [70], must continue to play in relaying information on the importance of antenatal care utilization, delivery at health facilities, and postnatal care, in preventing perinatal mortality.

The study was a large survey and national representative with analysis including regional variation, factors at individual and community level. The study had the following limitations: the study design was cross-sectional and assessed the information from the 5 years prior to the survey period, it was impossible to establish the cause and effect relationship. There might be recall bias, particularly for age or other retrospective data relying on memory of past events. However, DHS uses data from pregnancy, birth, and contraceptive calendar, and corresponding, and checking of the data was performed to minimize recall bias [19]. Medical and obstetric factors of the mother were not well addressed. The nature of data limits the inclusion of all possible factors that could affect perinatal mortality. Variables such as medical and obstetric factors of the mother were not well addressed. Moreover, the perinatal mortality was only captured after the death had occurred as the result difficult to determine the causes. DHS data use contraceptive calendar method to calculate stillbirth, but this has a limitation in that twins or triplets are recorded with a single code "B" [19] which might results in undercounting of stillbirths.

## Conclusion

In Ethiopia, the prevalence of perinatal mortality is high and had spatial variations across the country. Strengthening partner's education, family planning for longer birth interval, ANC, and delivery services are essential to reduce perinatal mortality and achieve sustainable development goals in Ethiopia. Disparities in perinatal mortality rates should direct efforts to address inequities in maternal and neonatal healthcare services all over the country.

## Supporting information

**S1 File.**
(DOCX)

## Acknowledgments

We would like to acknowledge CSA for allowing access to the 2016 EDHS data set. We also acknowledge Dr Tara Wilfong MD, MPH, University Orlando, Florida Area 64 connections, Global Health Consultant for USAID Feed the Future Innovation Lab for Livestock Systems. Currently a Fulbright Scholar in Ethiopia at Haramaya University for her priceless contribution by editing language of this manuscript.

## Author Contributions

**Conceptualization:** Tesfaye Assebe Yadeta, Bizatu Mengistu.

**Data curation:** Tesfaye Assebe Yadeta, Lemma Demissie Regassa.

**Formal analysis:** Tesfaye Assebe Yadeta, Bizatu Mengistu, Tesfaye Gobena, Lemma Demissie Regassa.

**Investigation:** Tesfaye Assebe Yadeta, Tesfaye Gobena.

**Methodology:** Tesfaye Assebe Yadeta, Bizatu Mengistu, Tesfaye Gobena, Lemma Demissie Regassa.

**Project administration:** Tesfaye Assebe Yadeta.

**Resources:** Tesfaye Assebe Yadeta.

**Software:** Tesfaye Assebe Yadeta, Lemma Demissie Regassa.

**Supervision:** Tesfaye Assebe Yadeta, Tesfaye Gobena, Lemma Demissie Regassa.

**Validation:** Tesfaye Assebe Yadeta, Bizatu Mengistu, Lemma Demissie Regassa.

**Visualization:** Tesfaye Assebe Yadeta, Tesfaye Gobena, Lemma Demissie Regassa.

**Writing – original draft:** Tesfaye Assebe Yadeta, Bizatu Mengistu, Tesfaye Gobena, Lemma Demissie Regassa.

**Writing – review & editing:** Tesfaye Assebe Yadeta, Bizatu Mengistu, Tesfaye Gobena, Lemma Demissie Regassa.

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
