## [Decision Letter · Decision Letter 0]

10 Jun 2020

PONE-D-20-04429

Spatial pattern of perinatal mortality and its determinants in Ethiopia: Data from Ethiopian Demographic and Health Survey 2016.

PLOS ONE

Dear Dr. Yadeta,

Thank you for submitting your manuscript to PLOS ONE. After careful consideration, we feel that it has merit but does not fully meet PLOS ONE’s publication criteria as it currently stands. Therefore, we invite you to submit a revised version of the manuscript that addresses the points raised during the review process.

We look forward to receiving your revised manuscript.

Kind regards,

Natasha McDonald

Associate Editor

PLOS ONE

Journal Requirements:

http://hubcymru.org.uk/images/user/HCA%20Lancet%20Every%20Newborn%20Series.pdf

http://jogh.org/documents/issue201702/jogh-07-020501.pdf

The text that needs to be addressed is in the Introduction section.

In your revision ensure you cite all your sources (including your own works), and quote or rephrase any duplicated text outside the methods section. Further consideration is dependent on these concerns being addressed.

6. We note you have included a table to which you do not refer in the text of your manuscript. Please ensure that you refer to Table 3 in your text; if accepted, production will need this reference to link the reader to the Table.

Reviewers' comments:

Reviewer's Responses to Questions

**Comments to the Author**

1. Is the manuscript technically sound, and do the data support the conclusions?

Reviewer #1: Partly

Reviewer #2: Yes

Reviewer #3: No

2. Has the statistical analysis been performed appropriately and rigorously? 

Reviewer #1: Yes

Reviewer #2: Yes

Reviewer #3: Yes

3. Have the authors made all data underlying the findings in their manuscript fully available?

Reviewer #1: Yes

Reviewer #2: Yes

Reviewer #3: Yes

4. Is the manuscript presented in an intelligible fashion and written in standard English?

Reviewer #1: Yes

Reviewer #2: No

Reviewer #3: No

5. Review Comments to the Author

Reviewer #1: This is a sound paper, with strong justification. Before publication, I would like the authors to improve some areas I have listed below:

1. Generally the English should be polished. There are many places throughout the manuscript. For example, line 27: insert ''the'';

line 45 -to be intensified

line 56- insert has

2. line 35: Be specific on the type of GIS software, is it ArcGIS?

3. lines 42-45: These lines is a repeat of the results section. There is no conclusion drawn here.

4. lines 63 to 67: please specify which of the citations 7 to 9 is identified for each determinant. Does it mean each of the 3 citations have all the determinants in them.

5. line 72: the citation at the end of that line should be 11.

6. line 154: The outcome should change to Y_ij so that i should be for the child and child-level factors, while j is for the community factors.

7. line 161: insert comma after rural, and before distance

8. Tables 1 and 2: These should be changed so that the last columns should report the percentage of children who died in pre-natal (row percentages). In other words, these tables should already inform the reader on the association between the outcome and risk factors. Therefore some of the text, in lines 215 to 229, should change to reflect these patterns.

9. Table 3 is not cited in the text. This should appear on lines 252 onwards.

10. Text on lines 282 to 312 should refer to a Table.

11. Line 399 mentions ''recall bias''. How was this resolved, or how does it affect the results.

Reviewer #2: This paper provides a useful investigation of the spatial pattern and determinants of perinatal mortality in Ethiopia. I’m not a geospatial expert, and it’s not clear exactly how geospatial techniques have been applied alongside standard random effects methods to adjust for and explore clustering. It would be useful if you can explain more clearly what the spatial analysis adds above a multivariable model with random effects to explore the determinants of PNM grouping factors related to individuals, households and communities. I general, the descriptions in the results and results tables don’t make it clear what is being included in each model. The manuscript could do with professional copy-editing in English, particularly the background and discussion sections. I have the following suggestions for revisions:

1. Abstract, lines 37-41 - Can you list separately those positively and negatively associated with PMR?

2. Abstract, lines 43-45 - This sentence repeats the one in the results in lines 38-41. No need to say this twice. IN the conclusion you should focus more on the implications.

3. Background, line 54 - This is not a very good opening sentence. How about "Perinatal mortality includes stillbirths and early neonatal deaths, and is an important indicator of..."

4. Background, lines 56-57 - This sentence doesn't read well.

5. Background, lines 63-67 - Where do these risks relate to? Need to start the sentence by saying something like "Multi-

country reviews found that ..., ... and ... were significantly associated with perinatal mortality risk."

6. Methods, line 92 - Hasn't there been a more recent census?

7. Methods, line 99 - Please mention which EDHS survey this refers to before stating the number of participants. I know the dates of data collection are given below, but we still need to know which survey you are referring to before you give the numbers.

8. Methods, lines 146-147 - Elsewhere DHS defines a stillbirth as "Number of foetal deaths in pregnancies of seven or more months", which means the same, but is a bit more specific in relation to the foetus.

9. Results, line 215 - You haven't defined the study period in the methods. Is this all pregnancies, or just in the last five years?

10. Results, lines 216-217 - Better to give median and range for age, as likely not to be normally distributed.

11. Tables - 1 d.p. would be sufficient, depending on journal's policy.

12. Results, line 224 - Again, median and range would be better.

13. Results, line 232 onwards - Before exploring the spatial distribution and factors associated with PNM, you need to present the overall PMR in this survey. Would also be good to show PMR for each variable/category in Table 2.

14. Results, line 258 - Could you call this post-secondary education to make it clearer? Usually this would be expressed as "83% lower", as 1 is the reference.

15. Results, line 259 - Likewise, this is "37% lower". Please check with all AORs below 1 that are reported in the text.

16. Table 3 - You need to explain the difference between Model 2, 3 and 4 in the headings or footnote to the table. Also where is model 1? Is that a crude model? Why not show that one as well?

17. Table 3 – Do all models have random effects for cluster? Or only models 3 and 4?

18. Table 3 – You could exclude some variables from multivariable models that are not associated with PNM in the crude model.

19. Table 3 - May be better to combine secondary and higher education into one group, to reduce the number of parameters in models and reduce data sparsity.

20. Results, line 288 - Was household-level clustering/random effects included in the model? Or these were just factors related to the household? I don't think you could include household-level clustering/random effects for PNM since there would be many households with only one pregnancy. So if it is not included as a random effect in the multi-level model, better to call these "household-related factors" to avoid confusion with levels in the model.

21. Table 4 - Models 2 and 3 seem to be the opposite way around to Table 3. I think Model 2 should be individual only and model 3 should be community only.

22. Discussion, line 353 - decreased by 41%.

23. Discussion, line 357 - Your data shows less than 15 months. If you want to show for less than 2 years you need to re-categorise this variable.

24. Discussion, lines 401-403 - I don't understand this sentence. The same person cannot have a stillbirth and a live birth in the same month unless they are twins and born at the same time.

25. Figure 2 - The colour scheme in the legend is not visible. Are the points shown on the map the sampled clusters from the survey? What do the different colours of the geographical regions mean?

26. Figure 3 - You need to explain LLR in full, or in a footnote so that this can be interpreted as a standalone figure. Or change the legend as this is not actually showing perinatal mortality rates, but likelihood ratios. So explain what is being compared with what.

Reviewer #3: The authors used data from the Ethiopia DHS to estimate risks of perinatal mortality at individual and community levels and to identify geographical clustering of perinatal mortality. While statistical analyses appear to be conducted correctly, I am not clear about how the authors defined the study subjects. (The issue may simply be with English writing.) I read (Line 99-100) that the authors included pregnant women, which suggests to me that the women were still carrying a foetus in utero? How do we know the ultimate survival status of their babies? We don’t – some of them will be born alive and some will be stillborn. If all the babies in utero of pregnant women were counted by the authors as a live birth who survive the first 7 days of their life, the authors would have underestimated the perinatal mortality rate. I couldn’t understand what the authors meant at line 402 where they discussed the possibility that the study may have undercounted stillbirths. Maybe I am pointing out the same issue. The authors are introducing unnecessary bias here.

Another issue is the English language. The text will benefit from English editing, the methods section is readable though.

Line 99-100: Already pointed out above but the description of the study participants is unclear possibly an issue with English writing but equally a technical issue introducing a bias). It can read as if women who were 7 month-pregnant at the time of the survey were included in the study but I don’t think that’s what the author meant? Or is it? It is not possible to count the number of stillbirths and neonatal deaths (the numerator of perinatal mortality rate) until after all the pregnancies have ended. So I don’t think “7320 women pregnant” (assuming that this means 7320 pregnant women) represent the study subjects?

However, having now read line 168 onward, it seems that the study included “pregnant women with 7 months and more”. How do we know that the babies from these women will live or die?

At line 138-139, the same information is repeated as above that 7320 women…were included. No need to repeat the same information.

Line 107: The same information as line 88 is repeated here.

Line 158: Abbreviations need to be spelt out at first appearance.

Results

It would have been more informative if the characteristics of women who had a perinatal death and who didn’t were compared, rather than describing the characteristics of the entire study subjects as in Table 1 and table 2.

Measure of variation and model fit statistics

I think the columns in table 4 need to be swapped/re-labelled. The text at line 288 reads that after adjusting the model for “household level factors (model II)”…., but in table4, I see that model 2 is adjusted for community level factors, which should actually be model 3?

Discussion

Linen 349- Can the authors provide some explanation for the relationship between older mothers and perinatal mortality? Dismissing it as “unclear” sounds sloppy.

The health extension program is discussed and it is suggested that “the findings of our study serve to further emphasize the important role that HEWs must continue to play in relaying information about perinatal death” but results of the study suggest the importance of institutional delivery. What role doe the HEWs play in institutional delivery? Could the authors comment?

Line 401- I don’t quite understand what the authors are trying to say ‘in case of a stillbirth and a live birh,,,’

6. PLOS authors have the option to publish the peer review history of their article (what does this mean?). If published, this will include your full peer review and any attached files.

Reviewer #1: No

Reviewer #2: No

Reviewer #3: Yes: Atsumi Hirose

---

## [Author Response · Author response to Decision Letter 0]

17 Jul 2020

I have attached as response for reviewer

---

## [Decision Letter · Decision Letter 1]

8 Oct 2020

PONE-D-20-04429R1

Spatial pattern of perinatal mortality and its determinants in Ethiopia: Data from Ethiopian Demographic and Health Survey 2016.

PLOS ONE

Dear Dr. Yadeta,

Thank you for submitting your manuscript to PLOS ONE. After careful consideration, we feel that it has merit but does not fully meet PLOS ONE’s publication criteria as it currently stands. Therefore, we invite you to submit a revised version of the manuscript that addresses the points raised during the review process.

Reviewers and I believe the authors responded to all comments and suggestions. There are some minor issues, listed below, and mostly gramamatical. The manuscript will benefit from formal editing. 

We look forward to receiving your revised manuscript.

Kind regards,

Bernardo Lanza Queiroz, Ph.D

Academic Editor

PLOS ONE

Reviewers' comments:

Reviewer's Responses to Questions

**Comments to the Author**

1. If the authors have adequately addressed your comments raised in a previous round of review and you feel that this manuscript is now acceptable for publication, you may indicate that here to bypass the “Comments to the Author” section, enter your conflict of interest statement in the “Confidential to Editor” section, and submit your "Accept" recommendation.

Reviewer #2: All comments have been addressed

Reviewer #3: (No Response)

2. Is the manuscript technically sound, and do the data support the conclusions?

Reviewer #2: Yes

Reviewer #3: Yes

3. Has the statistical analysis been performed appropriately and rigorously? 

Reviewer #2: Yes

Reviewer #3: Yes

4. Have the authors made all data underlying the findings in their manuscript fully available?

Reviewer #2: Yes

Reviewer #3: Yes

5. Is the manuscript presented in an intelligible fashion and written in standard English?

Reviewer #2: Yes

Reviewer #3: No

6. Review Comments to the Author

Reviewer #2: The authors have clearly responded to the suggestions made previously, and the English is improved. However, there are still a few areas where the language could be improved further. I’ve highlighted a few below, as well as a suggestion for Tables 1 and 2:

1. Abstract, line 31 – “…women who at delivered at seven or months…”

2. Abstract, line 45 – Shouldn’t this be for longer birth intervals?

3. Abstract, lines 47-48 – do you mean “Disparities in perinatal mortality rates should be addressed alongside efforts to address…”

4. Background, line 54 – is higher

5. Background, line 67 – delete “which”

6. Background, lines 71-79 – needs some copy editing

7. Discussion, line 349 – associated with reduced perinatal mortality

8. Conclusion, line 443 – Again, shouldn’t this be longer birth intervals?

9. Tables 1 and 2 – the addition of the perinatal mortality is useful in these tables, but this should be expressed as the PMR for each category, not the column percent adding up to 100%. For example, we want to know the perinatal mortality rate for rural and the perinatal mortality rate for urban so that we can see which is higher.

Reviewer #3: Thank you for addressing my comments.

There are still minor issues, mostly grammatical errors. I listed below some of them which can be corrected easily but there are more. The manuscript will probably benefit from proofreading / editing by a native English speaker.

Line 31: “7 days postpartum” is not needed.

Line 44: Not sure if it is correct to refer to mortality as prevalence. Could you consider changing this?

Line 45: Women’s education was not associated with perinatal mortality, so perhaps need to drop the mention of women’s education here.

Line 54: Perinatal mortality ‘has’ should be changed to ‘is’

Line 67: ‘which’ can be deleted.

Line 73: I think this sentence is awkward.

Line 98: can you add ‘of whom’ just before ‘7230 women….were used…’

Line 223: ‘live’ birth rather than ‘alive’ birth.

Line 225: Can you please check if the statistics for urban residence is correctly reported? I assume that 84% were rural residents rather than urban residents. I think Table 1 also needs to be corrected in relation to this.

Line 233-238: It would be much better if the order of presenting different factors correspond with the order they appear in table 2.

Line 247: Can you please check this statistics> Mortality was higher in urban than in rural area? This is different from what is reported in the abstract and also not in line with existing literature. I suspect this is a typo. Should have been higher in rural area’?

Line 249: I think Figure 1 is currently ‘Global Moran’s I test of cluster distribution for perinatal mortality in Ethiopia, from EDHS 2016’. Please check the figures and order them correctly.

Line 271: The sentence is awkward. Can drop ‘were’?

Line 363: “A possible explanation” rather than ‘A possible the explanation”

7. PLOS authors have the option to publish the peer review history of their article (what does this mean?). If published, this will include your full peer review and any attached files.

Reviewer #2: No

Reviewer #3: **Yes: **Atsumi Hirose

---

## [Editor Report · Decision Letter 2]

4 Nov 2020

Spatial pattern of perinatal mortality and its determinants in Ethiopia: Data from Ethiopian Demographic and Health Survey 2016.

PONE-D-20-04429R2

Dear Dr. Yadeta,

We’re pleased to inform you that your manuscript has been judged scientifically suitable for publication and will be formally accepted for publication once it meets all outstanding technical requirements.

Kind regards,

Bernardo Lanza Queiroz, Ph.D

Academic Editor

PLOS ONE
---

## [Editor Report · Acceptance letter]

10 Nov 2020

PONE-D-20-04429R2 

Spatial pattern of perinatal mortality and its determinants in Ethiopia: Data from Ethiopian Demographic and Health Survey 2016. 

Dear Dr. Yadeta:

I'm pleased to inform you that your manuscript has been deemed suitable for publication in PLOS ONE. Congratulations! Your manuscript is now with our production department. 

Kind regards, 

on behalf of

Dr. Bernardo Lanza Queiroz 

Academic Editor

PLOS ONE